# PEP2PROB BENCHMARK: PREDICTING FRAGMENT ION PROBABILITY FOR MS²-BASED PROTEOMICS

## ABSTRACT

Proteins perform nearly all cellular functions and constitute most drug targets, making their analysis fundamental to understanding human biology in health and disease. Tandem mass spectrometry (MS²) is the major analytical technique in proteomics that identifies peptides by ionizing them, fragmenting them, and using the resulting mass spectra to identify and quantify proteins in biological samples. In MS² analysis, peptide fragment ion probability prediction plays a critical role, enhancing the accuracy of peptide identification from MS² spectra as a complement to the intensity information. Current approaches rely on global statistics of fragmentation, which assumes that a fragment's probability is uniform across all peptides. Nevertheless, this assumption is oversimplified from a biochemical principle point of view and limits accurate prediction. To address this gap, we present **Pep2Prob**, the first comprehensive dataset and benchmark designed for peptide-specific fragment ion probability prediction. The proposed dataset contains fragment ion probability statistics for 608,780 unique precursors (each precursor is a pair of peptide sequence and charge state), summarized from more than 183 million high-quality, high-resolution, HCD MS² spectra with validated peptide assignments and fragmentation annotations. We establish baseline performance using simple statistical rules and learning-based methods, and find that models leveraging peptide-specific information significantly outperform previous methods using only global fragmentation statistics. Furthermore, performance across benchmark models with increasing capacities suggests that the peptide-fragmentation relationship exhibits complex nonlinearities requiring sophisticated machine learning approaches. Pep2Prob provides a standardized evaluation framework that will accelerate algorithmic innovation in computational proteomics while introducing a biologically significant prediction task to the machine learning community.

## 1 INTRODUCTION

Proteomics, the large-scale study of proteins, provides critical insights into cellular function, disease mechanisms, and potential therapeutic targets (Topol, 2024). Tandem mass spectrometry (MS²) has emerged as the predominant analytical technique for identifying and quantifying proteins in complex biological samples (Aslam et al., 2016). In MS²-based proteomics, proteins are first digested enzymatically into peptides, ionized and separated by their mass-to-charge ratio ($m/z$); then the selected peptide ions are fragmented, generating fragment ion spectra that serve as "fingerprints" for peptide identification.

The interpretation of MS² spectra represents a fundamental pattern recognition challenge where computational methods match observed spectral patterns to peptide sequences. Core approaches—database search (Kim & Pevzner, 2014; Tyanova et al., 2016), de novo sequencing (Frank & Pevzner, 2005; Tran et al., 2017; Yilmaz et al., 2024), and spectral library matching (Dorl et al., 2023; Shao et al., 2013; Wang et al., 2010)—all require accurate prediction of peptide fragmentation behavior as a critical prerequisite. Fragment ion

probability prediction (see Figure 1) serves as a key intermediate task that directly impacts several downstream applications, including peptide identification (Kong et al., 2017), post-translational modification (PTM) localization (Witze et al., 2007), and protein quantification (Pan et al., 2009). This probability provides complementary information to intensity modeling, effectively serving as a confidence weighting mechanism that distinguishes signal from noise in complex biological samples. For more related works, see Appendix B.

Current fragment ion prediction methods widely used in peptide identification workflows (Bandeira et al., 2008; Dančík et al., 1999; Kim & Pevzner, 2014) rely on global statistics that treat all peptides uniformly, assuming fragmentation probabilities depend only on a fragment's partial information, namely, fragment type (e.g., $b$-ions vs. $y$-ions) and charge state. Such approaches completely disregard peptide-specific features, similar to predicting natural language tokens using only part-of-speech tags without considering context. In reality, peptide fragmentation is governed by complex sequence-dependent factors: neighboring amino acids influence bond stability, charge mobility varies with sequence composition, and local chemical environments create position-specific effects.

To address these limitations, we introduce **Pep2Prob**, the first large-scale dataset together with a comprehensive benchmark specifically designed for peptide-specific fragment ion probability prediction. Pep2Prob comprises fragment probability statistics for 608,780 unique precursors, derived from over 183 million high-resolution higher-energy collisional dissociation (HCD) MS$^2$ spectra with validated peptide assignments. We establish comprehensive benchmarks by systematically evaluating the ability of various baseline models, including simple statistical models and large neural networks, showing that increasing the model's capacity can help capture complex peptide-specific fragmentation patterns. Our empirical analysis reveals two key findings:

1. Incorporating peptide sequence information yields substantial performance improvements over global statistics. In particular, a simple empirical statistical method incorporating only fragment sequence information achieves $\approx 0.18$ test set L1 loss compared to $\approx 0.24$ from conventional global modeling methods using only ion type and charge information.
2. After including peptide-specific information, we observe that prediction accuracy continuously improves with increasing model capacity: test set L1 losses decrease from 0.126 (linear regression) to 0.069 (neural network) to 0.056 (transformer). This trend indicates that the relationship between peptide sequences and fragmentation probabilities exhibits complex nonlinearities that simple models fail to capture. These results demonstrate that effective fragment ion prediction requires sophisticated machine learning (ML) approaches.

Pep2Prob provides a standardized framework for the proteomics and ML communities to advance fragment ion prediction. We anticipate that the complex sequence-to-fragmentation relationships captured in our dataset will inspire new ML algorithms, yielding immediate practical benefits for downstream proteomics applications, including peptide identification, PTM localization, and biomarker discovery.

## 2 PROBLEM FORMULATION, TASK DEFINITION AND NOTATIONS

A peptide is a short chain of amino acids linked by peptide bonds, serving as the fundamental unit of protein identification in proteomics. In MS$^2$, peptides are first ionized and isolated as precursor ions, which are specific peptides with defined charge states:

$$\underline{p}\text{recursor} = (\text{peptide } \underline{seq}\text{uence}, \underline{c}\text{harge state}) \quad \text{e.g.} \quad (\text{PEPTIDE}, 2+). \tag{2.1}$$

In the example above, each letter in 'PEPTIDE' represents an amino acid, and 2+ indicates the precursor has 2 positive charges. These precursors are then fragmented through collisional activation, breaking peptide bonds to produce characteristic fragment ions. The resulting MS$^2$ spectrum, a collection of fragment ion peaks at specific mass-to-charge ratios, serves as a molecular fingerprint that enables peptide identification.

Figure 1: **(A)** Illustration of the task of fragment ion probability prediction, aiming to estimate the likelihood that each potential fragment ion will be observed in the MS$^2$ spectrum of a precursor. **(B)** Pep2Prob establishes the *first* machine learning dataset and benchmark for peptide-specific fragment ion probability prediction, with rich applications in proteomics studies.

As illustrated in Figure 1(A), fragment ions are the products of peptide fragmentation, each uniquely specified by three key attributes:

$$\underline{f}\text{ragment ion} = (\text{ion }\underline{t}\text{ype, }\underline{c}\text{harge, position }\underline{n}\text{umber}) \tag{2.2}$$

The ion type $t$ indicates which portion of the original peptide is retained: $a$- and $b$-ions contain the prefix, while $y$-ions contain the suffix. Here, $a$- and $b$-ions further distinguish the chemical structure of the prefix fragment, where $a$-ions result from the loss of CO from corresponding $b$-ions ($a$-ion is only possible when $c = +1$ and $n = 2$). Note that it is commonly assumed that each fragment is produced by a single cut; hence, each fragment is either a prefix or a suffix. Next, the charge state $c$ specifies how many positive charges the fragment carries. Finally, the position number $n$ denotes where the peptide bond was cleaved, counted from the respective end. For instance, $(b, 2+, 3)$ denotes a doubly-charged $b$-ion fragment containing the first three amino acids, and $(y, 1+, 5)$ denotes a singly-charged $y$-ion fragment containing the last five amino acids.

During MS$^2$ analysis, fragmented ions are then separated by their mass-to-charge ratios and detected to generate a spectrum. Formally, a spectrum is a set of peaks:

$$S = \{s_1, s_2, ...\} \quad s_i = (m/z_i, I_i) \tag{2.3}$$

where each element is a peak defined by its mass-to-charge ratio ($m/z_i$) and intensity ($I_i$). Each observed peak originates from a specific fragment ion of the precursor, while the intensity quantifies how frequently that particular fragmentation occurs.

Not all theoretically possible fragments appear as peaks in the spectrum, and only a small subset generates detectable signals. This could be due to differences in the stability of the peptide bond, the preferences of amino acid-specific fragmentation, and instrument detection limits. The task of **fragment ion probability prediction** aims to estimate the likelihood that each potential fragment will be observed in the MS$^2$ spectrum of a precursor. We mathematically denote this likelihood as:

$$\mathbb{P}(f|p) \overset{\text{def}}{=} \mathbb{P}(\text{fragment ion } f \text{ shows up in the precursor } p\text{'s spectrum}) \tag{2.4}$$

The quantity serves as complementary information to intensity values, enhancing peptide identification algorithms by distinguishing likely fragments from theoretical possibilities.

Existing methods (Kim & Pevzner, 2014; Bandeira et al., 2008; Dančík et al., 1999) for modeling $\mathbb{P}(f|p)$ only take the fragment ion $f$ as input and ignore all precursor $p$ information. This amounts to assuming that $\mathbb{P}(f|p)$ is uniform across all precursors/peptides, which is unreasonable from a biochemical perspective: different peptide sequences exhibit distinct fragmentation patterns due to varying amino acid properties and bond stabilities. Pep2Prob aims to demonstrate that $\mathbb{P}(f|p)$ has a nontrivial dependency on precursor information (peptide sequence and charge state), and that incorporating this information in the prediction of fragment ion probability improves performance significantly.

## 3 PEP2PROB DATASET CONSTRUCTION

Pep2Prob is built upon 227 human HCD mass spectrometry datasets and their associated peptide-spectrum matches, which are publicly accessible via the Mass Spectrometry Interactive Virtual Environment (**MassIVE**) repository (http://massive.ucsd.edu) (MassIVE, 2025). We also leverage the MassIVE Knowledge Base (**MassIVE-KB**) in vivo library (version 2.0.15) (Wang et al., 2018), which was curated from these HCD mass spectrometry datasets, for precursor selection. The MassIVE-KB library contains a reference spectrum for each of the 5,948,126 precursors with a global precursor false discovery rate estimated at 0.1%.

Precursors from MassIVE-KB were filtered based on the following criteria: (1) peptide sequence lengths between 7 and 40 residues, (2) absence of modifications, and (3) at least 10 associated spectra. This rigorous filtering process resulted in a curated dataset of 183,263,674 spectra corresponding to 608,780 unique precursors. The distributions of precursor counts across sequence lengths and charge states are summarized in Figure C.1 in the Appendix.

From this refined spectral dataset, we construct the Pep2Prob dataset as briefly described below and detailed in subsequent sections. We annotate each precursor's associated spectra and aggregate these annotations to calculate occurrence probabilities for all possible fragment ions for each precursor. Furthermore, we implement a novel train/test splitting strategy specifically designed to prevent structurally similar peptides from appearing in both training and test sets.

### 3.1 MASS SPECTRUM ANNOTATION AND FEATURE REPRESENTATION

Mass spectrum annotation is the process of assigning one theoretically possible fragment ion to each observed peak in an MS$^2$ spectrum. Each annotated fragment ion is a tuple defined in (2.2).

For each precursor $p$ in the Pep2Prob dataset, the peptide sequence length ranges from $N_p \in [7, 40]$ and the charge state from $C \in [1, 8]$ (see Appendix Figure C.1). We now define the annotation space—the set of theoretically possible fragment ions across all precursors in the dataset. Although precursor charge states may reach up to 8, we only consider fragment ions with charges 1, 2, or 3 when constructing the annotation space, as higher-charged fragments are rare and less reliably observed in high-resolution HCD spectra. Let $c_{\max} = 3$ and $N_{\max} = 40$. The annotation space $\mathcal{F} = \{f = (t, c, n)\}$ consists of:

1. One entry for the $a$-ion with +1 charge, which is always included,

2. $c_{\max}(N_{\max} - 1)$ entries for $b$-ions, spanning charges 1 to $c_{\max}$ and positions $j \in \{1, \ldots, N_{\max} - 1\}$,

3. Similarly, $c_{\max}(N_{\max} - 1)$ entries for $y$-ions with the same possible charge and position combinations as $b$-ions.

The annotation space, therefore, has a total size $d = 235$.

Each observed $MS^2$ spectrum is a list of peaks, denoted as $S = \{(MZ_i, I_i) : 1 \leq i \leq d'\}$, where $d'$ is the number of peaks observed in the spectrum, $MZ_i$ is the mass-to-charge ratio ($m/z$) of the $i$-th peak, and $I_i$ is its intensity. The number $d'$ varies across spectra and is typically much smaller than $d$. To annotate the spectrum, we compute the theoretical $m/z$ value for each of the $d$ fragment ions in $\mathcal{F}$ and match each observed peak $S$ within a predefined tolerance window $\delta = 0.05$ Th, i.e., $|MZ_i - m/z| \leq \delta$, to the most possible fragment ion (see details in Appendix C.3). If a match is found, the intensity $I_i$ is recorded; otherwise, the corresponding entry is set to zero. This produces a length-$d$ intensity vector for each precursor-spectrum pair. Stacking these vectors across all precursors yields a 2D matrix where: Each row corresponds to a precursor-spectrum pair; Each column corresponds to a fragment ion $f = (t, c, n)$; Each entry is either a matched intensity from the spectrum or zero.

**Fragment ion mask.** While the annotation space has a fixed dimension $d = 235$, not all fragment ions are valid for each precursor $p$ due to sequence length and charge state constraints. To account for this, we define a binary *ion mask* $\pi(f, p) \in \{0, 1\}$ that indicates whether fragment ion $f = (t, c, n)$ is theoretically possible for a given precursor $p$ with sequence length $N_p$ and charge state $C$. The entries of the mask are defined by: for $f \in \mathcal{F}$,

$$\pi(f, p) \stackrel{\text{def}}{=} \begin{cases} 1, & \text{if } c \leq \min\{C, 3\} \text{ and } n < N_p, \text{ or } t = a, \\ 0, & \text{otherwise.} \end{cases} \tag{3.1}$$

This mask is applied during both training and evaluation. It ensures that predictions and loss computations are restricted to fragment ions that are chemically valid for the given precursor. This reduces the dimensionality of the output space and improves training efficiency and model interpretability.

## 3.2 FRAGMENT ION PROBABILITY STATISTICS

To estimate the statistical likelihood of fragment ion appearances, we construct the fragment ion probability dataset based on repeated $MS^2$ measurements for the same precursor. Suppose a given precursor $p$ is observed in $D$ distinct $MS^2$ spectra. Each spectrum is first annotated and normalized to form a non-negative intensity vector $\boldsymbol{I}^{(i)} = \{I_f^{(i)}\}_{f \in \mathcal{F}}$, where $I_f^{(i)}$ is the intensity for fragment $f$ and the $\ell_1$ norm $\|\boldsymbol{I}^{(i)}\|_1 = 1$ for all $i \in \{1, \ldots, D\}$. We then estimate the probability that each fragment ion appears across the $D$ spectra. We use $\epsilon = 10^{-6}$ as the threshold for fragment presence. The empirical probability that fragment ion $f \in \mathcal{F}$ appears is computed by

$$\mathbb{P}(f|p) = \frac{1}{D} \sum_{i=1}^{D} \mathbb{1}\{I_f^{(i)} > \epsilon\}. \tag{3.2}$$

If the computed value is below 0.001, we set it to zero to suppress unreliable noise. This results in an ion fragment probability vector for precursor $p$: $\boldsymbol{P}_p = \{\mathbb{P}(f|p) : f \in \mathcal{F}\} \in [0, 1]^d$. Stacking these probability vectors across all precursors $p$ yields our Pep2Prob dataset.

**Learning objective.** Our goal is to train machine learning models that take a precursor $p$ as input and predict the corresponding ion probability vector $\boldsymbol{P}_p$. Denote the model prediction for precursor $p$ by

$$\hat{\boldsymbol{P}}_p = \{\hat{\mathbb{P}}(f|p) : f \in \mathcal{F}\} \in [0, 1]^d.$$

During training and evaluation, the ion mask $\pi$ defined by (3.1) is applied to both the predicted and ground-truth probability vectors to restrict computations to valid fragment ion dimensions.

## 3.3 TRAIN AND TEST SPLIT

Many precursors in our dataset share common sequence patterns, particularly long prefixes or suffixes — e.g., "ABCDEFGHIJK", "ADCDEFGHIJK", "ABCCDEFGHIJK", and "EFGHIJK". It is reasonable to *not* split

such similar precursors into train and test datasets separately, otherwise it may lead to data leakage (Joeres et al., 2025) and overoptimistic evaluation (Hobohm et al., 1992; Sander & Schneider, 1991; Teufel et al., 2023) due to similar sequences appearing in both training and test sets. If similar sequences are split across the training and test sets, the model may easily generalize by memorizing patterns seen during training.

We construct an undirected graph where each node represents a precursor. An edge is added between two nodes if their peptide sequences satisfy: **Identical** — they are the same; **SharePrefix** or **ShareSuffix** — they share a common prefix or suffix of length 6. If none of these conditions are met, the pair is labeled as having **NoConnection**, and no edge is added. Given that the minimum peptide sequence length is 7 (Figure C.1 in the Appendix), this criterion effectively captures meaningful local similarities without being overly inclusive.

The resulting graph decomposes into disconnected components, each representing a group of similar sequences. To create train/test splits, we sort these components by size and distribute them into five balanced folds using a greedy strategy: each new component is assigned to the currently smallest fold. One fold is used for testing, and the remaining four for training. This component-based split ensures no shared local motifs between training and test sets, leading to more realistic and robust evaluation of the model's generalization ability.

## 4 BENCHMARKING FRAGMENT ION PROBABILITY PREDICTION

### 4.1 BASELINE MODELS AND EXPERIMENTAL SETUP

We benchmark a set of methods on the Pep2Prob dataset with two goals: (1) to establish baseline performance metrics; (2) to determine how strongly the fragment-ion probability $\mathbb{P}(f|p)$ depends on $p$ and how complex that dependency is, thereby gauging the model capacity needed for accurate fragment-ion prediction.

We first apply two statistical-rule-based approaches: one global method that ignores precursor information entirely and one sequence-aware method that partially incorporates precursor context. Additionally, we implement three machine learning baselines—linear regression, neural networks, and transformers—with varying capacities to model complex relationships. Each ML model incorporates the fragment-ion mask $\pi(f, p)$ in (3.1) and is trained to minimize L1 loss defined as:

$$\ell_1(\boldsymbol{P}_p, \hat{\boldsymbol{P}}_p) \stackrel{\text{def}}{=} \frac{\sum_{f \in \mathcal{F}} \left| \mathbb{P}(f|p) - \hat{\mathbb{P}}(f|p) \right| \cdot \pi(f, p)}{\sum_{f \in \mathcal{F}} \pi(f, p)}. \tag{4.1}$$

**Global Modeling (Global).** Our initial baseline computes fragment ion probabilities solely based on ion type $t$ and charge $c$ information, excluding $n$ and $p$. This predictor assumes $\mathbb{P}(f|p)$ is independent of the peptide sequence and the position number (Dančík et al., 1999). It is widely used in current peptide identification methods via database search and de novo sequencing, such as Bandeira et al. (2008) and MSGF+ (Kim & Pevzner, 2014). The predictor is built upon global statistics aggregated across precursors in the training set. For each unique fragment ion $f$, the model estimates:

$$\hat{\mathbb{P}}_{\text{Global}}(f = (t, c, n)) = \frac{\sum_{p \in \mathcal{D}_{\text{train}}} \sum_{f' = (t', c', n')} \pi(f', p) \cdot u(p) \cdot \mathbb{1}(t = t', c = c') \cdot \mathbb{P}(f'|p)}{\sum_{p \in \mathcal{D}_{\text{train}}} \sum_{f' = (t', c', n')} \pi(f', p) \cdot u(p) \cdot \mathbb{1}(t = t', c = c')} \tag{4.2}$$

where $u(p)$ is the number of spectra associated with precursor $p$, and $\pi(f, p)$ is the indicator function that equals 1 if precursor $p$ can theoretically produce fragment ion $f$, and 0 otherwise.

**Bag-of-Fragment-Ion (BoF).** This predictor extends global modeling by incorporating minimal precursor information. It computes fragment ion probabilities based on both the fragment $f$ and the fragment's amino acid sequence, which is determined jointly by $f$ (specifically its ion type and position number) and $p$. We denote this amino acid sequence as $\xi(f, p)$. For example, if $p$'s peptide sequence is 'ABCDE' and $f = (b, 1+, 2)$, then $\xi(f, p) = $ 'AB'; if $f = (y, 1+, 3)$, then $\xi(f, p) = $ 'CDE'.

The model's prediction rule is:

$$\hat{\mathbb{P}}_{\mathrm{BoF}}(f|p) = \frac{\sum_{p' \in \mathcal{D}_{\mathrm{train}}} \pi(f, p') \cdot u(p') \cdot \mathbb{1}(\xi(f, p) = \xi(f, p')) \cdot \mathbb{P}(f|p')}{\sum_{p' \in \mathcal{D}_{\mathrm{train}}} \pi(f, p') \cdot u(p') \cdot \mathbb{1}(\xi(f, p) = \xi(f, p'))} \tag{4.3}$$

This estimator conditions on fragments having identical amino acid sequences, capturing sequence-specific fragmentation patterns while remaining computationally tractable.

We then apply three standard baselines: (1) a **linear regression model (LR)**, which is an independent linear regressor for each fragment ion $f$; (2) a **1D Convolutional neural network (ResCNN-1D)** with residual connection, pooling, and batch normalization (He et al., 2016); and (3) a **fully-connected feed-forward neural network with residual connections (ResFFNN)**, which contains fully-connected feedforward layers and residual connections. The complete descriptions of these three models and their experimental configurations are in Appendix D. To establish these baselines, we need to first *encode* each precursor $p$ into a fixed-length one-hot vector $\mathbf{x}_p$: $\mathbf{x}_p = \mathbf{x}_c \oplus \mathbf{x}_{seq}$. which concatenates two components: a one-hot encoding of the charge state $\mathbf{x}_c$ and a one-hot encoding of the peptide sequence $\mathbf{x}_{seq}$, where the latter is formed by concatenating one-hot vectors for each amino acid position.

**Transformer.** The Transformer baseline replaces the ResFFNN with a stack of standard self-attention layers to capture longer-range dependencies in the precursor input. Using the same token-encoding of the precursor $p$ (amino-acid tokens plus charge tokens) followed by an output token, we train a Transformer (Vaswani et al., 2017) to output $\hat{\mathbb{P}}_{\mathrm{TF}}(f \mid p)$ for all fragment indices $f \in \mathcal{F}$ in one shot. Let $m = N_p + 1$ be the combined sequence length, where $N_p$ is the length of the peptide sequence. The token embedding is defined as $\mathbf{X}^{(0)} = \mathrm{Embed}(p) + P \in \mathbb{R}^{m \times d_0}$ where $P$ is a positional embedding and $\mathrm{Embed}$ is the embedding layer for each precursor. For more details of the architecture of the transformer model, see Appendix D.

Additional training details and computational resources for our models[1] are described in Appendix E.

## 4.2 EVALUATION PIPELINE

**Type 1: Norm-based metrics for probability values.** We first use norm-based metrics for comparing predicted and true fragment ion probability vectors. Common choices include mean squared error (MSE), which penalizes large deviations; L1 loss, which is more robust to outliers; and normalized spectral angle (SA) (Toprak et al., 2014), a scale-invariant metric measuring angular similarity. Higher SA values indicate better alignment between predicted and true probability vectors.

**Type 2: Support-recovery metrics for fragment ion existence[2].** To evaluate how well the predicted ion fragment probabilities capture true fragment presence, we compare the supports of the true and predicted probability vectors. For each precursor, we consider an ion as present if its true probability is nonzero and as predicted present if its predicted probability exceeds a threshold $\tau$. Given the precursor, we consider the support of the true probability vector $\boldsymbol{P}_p$ and the predicted probability vector $\hat{\boldsymbol{P}}_p$ as $S = \{f \in \mathcal{F} : \mathbb{P}(f|p) > 0\}, \hat{S} = \{f \in \mathcal{F} : \hat{\mathbb{P}}(f|p) > \tau\}$, where we use $\tau = 0.001$. Based on this, we compute standard classification metrics between $S$ and $\hat{S}$: **accuracy** (the fraction of correct predictions), **sensitivity** (the proportion of true fragments correctly identified), and **specificity** (the proportion of non-fragments correctly excluded).

**Two-level evaluation.** Each evaluation metric is computed at two levels: Precursor level, where we calculate the metric for each precursor and then average over all precursors, yielding an overall score per precursor; Fragment ion level, where we compute the metric for each individual ion and average over all ions in the dataset, providing a score per ion. The complete definitions of all the metrics are in Appendices F.1 and F.2.

---

[1] Code and dataset to reproduce all experimental results will be released publicly upon acceptance.

[2] Observe that predicting the fragment's existence is a straightforward downstream task from our predictions of its fragment ion probabilities. Previous works used the fragment ion intensity predictor to estimate its existence.

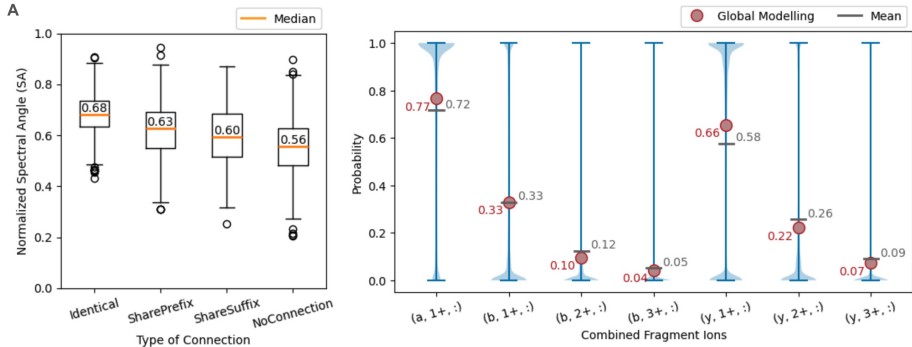

Figure 2: (A) The box plot of the normalized spectral angle (SA) based on the intersected fragment ion probabilities between precursors paired by **Identical**, **SharePrefix**, **ShareSuffix**, and **NoConnection**, defined in Section 3.3. (B) The probability distribution of fragment ions combined for all peptide sequences and fragmentation positions is shown for Pep2Prob in blue (mean value in gray) and for Global Modeling in red.

**Threshold for model outputs.** Consistent with our approach in constructing the Pep2Prob dataset, we apply a threshold to the model outputs during evaluation. Specifically, we set prediction values below 0.001 to zero, which maintains ion fragment probability at 99% precision. More generally, an optimal threshold $\tau$ could be determined for each precursor to recover the support of ion probability with desired precision-recall characteristics.

## 5 RESULTS AND ANALYSIS

**Data split based on sequence patterns.** Following the proposed training-test split in Section 3.3, we found that 339,102 precursors (55.7%) share identical peptide sequences and are therefore connected. Additionally, 429,863 precursors (70.6%) and 480,023 precursors (78.8%) share prefixes and suffixes of length 6 with others, respectively. Finally, we identified 204,716 isolated graph components, which are divided into 5 sets, each containing approximately 132,259 precursors from a total of around 58,000 precursors. Figure 2(A) shows that two precursors sharing common sequence patterns tend to have similar probabilities of the fragment ion occurring in both precursors, which validates the motivation for our training-test split strategy in Section 3.3.

**Distributional comparison on fragment ion probabilities.** Furthermore, Figure 2(B) illustrates the distribution of fragment ion probabilities across different ion types and charges in our dataset. We compare the Global Modeling approach (in red) to the empirical distribution of Pep2Prob (in blue). Notably, Global Modeling tends to overestimate or underestimate the probabilities of certain ion types. These discrepancies underscore the necessity of incorporating precursor-specific information to improve the accuracy of fragment ion probability estimation.

**Baseline comparison on fragment ion probability.** The left column of Table 1 shows Type 1 evaluations in Section 4.2. We see a clear hierarchy: the Global baseline yields the highest error and lowest SA similarity, indicating its inability to capture sequence-specific fragmentation patterns. BoF and LR progressively reduce prediction error, but only modestly. Deep architectures, ResFFNN and self-attention–based transformer, have higher model capacity, yielding lower error and higher SA for fragment ion probabilities. This progression underscores that the relation between fragmentation probabilities and precursor is complex and requires high-capacity ML models to learn effectively.

**Baseline comparison on fragment ion existence.** The right column of Table 1 evaluates the models' predictions on the existence of fragment ions (Type 2 evaluation). The global model achieves maximal

Table 1: Model performance on fragmentation ion probability prediction at precursor level. Best results are in **bold**, second-best ones are underlined. Analogous trends hold for fragment ion level in Tables G.1 and G.2 in the Appendix.

| Model | Type 1 Evaluation for Probability Values | | | Type 2 Evaluation for fragment ion existence | | |
|---|---|---|---|---|---|---|
| | **L1** | **MSE** | **SA** | **Acc** | **Sen** | **Spec** |
| Global | $0.2437 \pm 0.0002$ | $0.0994 \pm 0.0002$ | $0.5578 \pm 0.0004$ | $0.6993 \pm 0.0007$ | $\mathbf{1.0000 \pm 0.0000}$ | $0.0000 \pm 0.0000$ |
| BoF | $0.1788 \pm 0.0001$ | $0.1184 \pm 0.0001$ | $0.5086 \pm 0.0006$ | $0.8027 \pm 0.0008$ | $0.4435 \pm 0.0009$ | $\underline{0.7683 \pm 0.0005}$ |
| LR | $0.1258 \pm 0.0002$ | $0.0540 \pm 0.0002$ | $0.6951 \pm 0.0004$ | $0.7661 \pm 0.0053$ | $\underline{0.9213 \pm 0.0021}$ | $0.3771 \pm 0.0286$ |
| ResCNN-1D | $0.0720 \pm 0.0007$ | $0.0229 \pm 0.0005$ | $0.8079 \pm 0.0022$ | $0.8696 \pm 0.0023$ | $0.8640 \pm 0.0047$ | $0.7167 \pm 0.0069$ |
| ResFFNN | $0.0687 \pm 0.0002$ | $0.0213 \pm 0.0000$ | $0.8182 \pm 0.0003$ | $0.8708 \pm 0.0017$ | $0.8766 \pm 0.0026$ | $0.7150 \pm 0.0038$ |
| Transformer | $\mathbf{0.0558 \pm 0.0002}$ | $\mathbf{0.0163 \pm 0.0000}$ | $\mathbf{0.8467 \pm 0.0002}$ | $\mathbf{0.9514 \pm 0.0006}$ | $0.8001 \pm 0.0034$ | $\mathbf{0.9189 \pm 0.0017}$ |

Sensitivity at the expense of Specificity because it always assumes all theoretically possible fragment ions exist. BoF and LR already achieve high specificity and sensitivity, respectively. The ResFFNN further refines this balance by learning local sequence contexts, improving both true-positive and true-negative rates. Transformer achieves the highest overall accuracy and specificity, albeit with a modest drop in sensitivity. In addition, we found that model predictions must adequately account for the variance in possible fragment ions for each precursor input (Appendix G.2). These results illustrate that sophisticated ML models are essential not only for precise probability estimates but also for reliable fragment-existence predictions.

**Discussion on related tasks.** Fragment ion intensity or mass spectra prediction is a closely related task that plays an important role in proteomics pipelines. Although intensity provides complementary information about peptide fragmentation, it also indicates ion existence. However, we found that two state-of-the-art fragment ion intensity predictors, Prosit (Wilhelm et al., 2021) and AlphaPeptDeep (Zeng et al., 2022), achieved higher accuracy than BoF but performed much worse than Transformer (experimental details and results are in Appendix G.3). Additionally, database search is the primary method for peptide identification, while current popular tools (Bandeira et al., 2008; Kim & Pevzner, 2014) rely on fragment ion probabilities predicted by the global model. Pep2Prob enables modeling of precursor-specific fragmentation, helping improve database search design through more accurate fragment ion probability prediction models.

## 6 CONCLUSION AND LIMITATIONS

In summary, we have presented Pep2Prob, the first comprehensive dataset and benchmark designed specifically for peptide-specific fragment ion probability prediction in $MS^2$-based proteomics. Our benchmark experiments demonstrate two key findings: first, incorporating peptide sequence information significantly improves prediction accuracy compared to global approaches; second, the relationship between peptide sequences and fragmentation patterns exhibits intricate nonlinearities requiring sophisticated ML approaches to learn complex features efficiently.

One limitation of Pep2Prob is that it only captures marginal distributions of fragment ions conditioned on precursors, lacking the modeling of correlations between ion occurrences. Incorporating these correlations could further improve fragmentation modeling; however, constructing a dataset for this purpose would require parsing a large number of spectra for each precursor, which would significantly increase computational and storage requirements. Beyond this, Pep2Prob has two additional limitations regarding data source diversity. First, it excludes post-translational modifications (PTMs), which significantly alter fragmentation patterns, due to challenges in confidently identifying modified peptides at scale. Second, it contains only higher-energy collisional dissociation (HCD) spectra from Orbitrap instruments, not covering alternative fragmentation methods (ETD, CID) or different instrument platforms. Future work should address these limitations by incorporating collision energy, modified peptides, and expanding coverage across diverse fragmentation methods and instrument types, enhancing the practical utility of fragment ion prediction in proteomics workflows.

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

# Technical Appendices and Supplementary Material

## A    THE USE OF LARGE LANGUAGE MODELS

In the completion of this paper, we used large language models only for language polishing (grammar, clarity), typo correction, and minor code debugging. LLMs did not contribute to research ideation, methodological design, analysis, or drafting substantive content; all ideas, experiments, and results are our own.

## B    ADDITIONAL RELATED WORK

**ML for MS$^2$-based Proteomics.** The application of machine learning to MS$^2$ data for proteomic studies has evolved rapidly in recent years, with significant advances in (1) predicting peptide sequence from MS$^2$ spectra: de novo peptide sequencing (Qiao et al., 2021; Tran et al., 2017; Yilmaz et al., 2022; 2024) and (2) predicting properties in MS$^2$ of peptide sequence: fragment ion intensity prediction (Ekvall et al., 2022; Gessulat et al., 2019; Shouman et al., 2022; Zeng et al., 2022), retention time prediction (Bouwmeester et al., 2021; Gessulat et al., 2019; Zeng et al., 2022), and post-translational modification site prediction (Pokharel et al., 2022). Researchers have explored a diverse range of machine learning methods for these tasks, from gradient tree boosting algorithms (Degroeve et al., 2015; Degroeve & Martens, 2013; Gabriels et al., 2019) to more sophisticated models, including convolutional neural networks in (Liu et al., 2020), recurrent neural networks in (Tiwary et al., 2019), transformer architectures in (Ekvall et al., 2022; Yilmaz et al., 2022), and few-shot learning frameworks in (Tarn & Zeng, 2021). Recent advances have further enhanced performance by incorporating embeddings from protein language models (Hou et al., 2023; Rao et al., 2019; Weissenow et al., 2022). However, fragment ion probability prediction still relies on simple global modeling (Dančík et al., 1999) that ignores important peptide-specific fragmentation patterns but is widely used in popular peptide identification pipelines, such as MSGF+ (Kim & Pevzner, 2014).

As a remark, the HCD-only scope follows established best practices in MS-based proteomics method development, where leading approaches consistently focus on specific fragmentation types. Examples include Prosit series for ion intensity prediction (HCD model by Gessulat et al. (2019), CID model by Wilhelm et al. (2021), TMT model by Gabriel et al. (2022), and timsTOF model by Adams et al. (2024)), and also the leading de novo sequencing models: DeepNovo (HCD) (Tran et al., 2017) and Casanovo (HCD) (Yilmaz et al., 2022). This specialization is scientifically justified since different fragment types (HCD, CID, and ETD) produce fundamentally different ion types ($b/y$ v.s. $a/b$ v.s. $c/z$ ions), requiring distinct modeling approaches. HCD is a natural first step since it represents 90% of contemporary experiments and provides the highest quality data for reliable benchmark construction. Our work follows this proven development pattern and is extensible to other fragmentation types when equivalent datasets become available.

**MS$^2$ Datasets for ML in Proteomics.** While public mass spectrometry (MS) data repositories like PRIDE (Perez-Riverol et al., 2025) and MassIVE (MassIVE, 2025) host vast MS$^2$ data collections, they present challenges for applying machine learning directly due to variable quality and unidentified spectra. More structured resources have emerged for specific tasks recently, e.g., the nine-species dataset proposed in (Tran et al., 2017) for de novo sequencing and the PROSPECT serious datasets (Gabriel et al., 2024; Shouman et al., 2022) for predicting fragment ion intensity and retention time. The construction of these resources relies on the identification of mass spectra in public MS repositories. For instance, MassIVE-KB (Wang et al., 2018) addresses quality concerns by consolidating the best-representative spectra for each precursor across multiple experiments, thereby enhancing the reliability of model training (Yilmaz et al., 2024).

## C DATASET

### C.1 PRECURSOR

In Figure C.1, we summarize the statistics of the peptide sequence length of the precursors and the charge states of the precursors. Recall that for each precursor $p$ in the Pep2Prob dataset, the length of the peptide sequence is between 7 and 40, and the charge state could be an integer in $\{1, \ldots, 8\}$. Although precursor's charge states may go up to 8, we only consider fragment ions with charges 1, 2, or 3 when constructing the annotation space. In panel A of Figure C.1, precursor counts form a unimodal length distribution peaking at 11–12 ($\approx$50k each), and becoming rare past length 30. In panel B of Figure C.1 (log-scale y-axis), most precursors are charge states of 2+ or 3+.

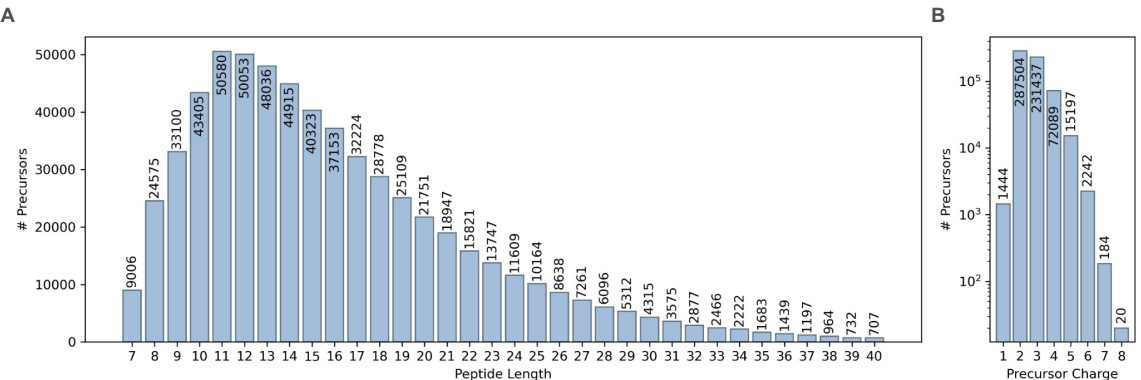

Figure C.1: The distribution of precursor counts across (**A**) sequence lengths and (**B**) charge state.

### C.2 FRAGMENT IONS

In Section 3.1, we define an annotation space of 235 fragment ions. Each fragment ion is a triple of (ion type, charge, position number). Here is the list of all considered fragment ions:

- 1 a-ion: $[('a', 1, 2)]$,
- 117 b-ions: $[('b', c, n) \ \forall \ c \in [1, 2, 3], n \in [1, 2, \ldots, 39]]$, and
- 117 y-ions: $[('y', c, n) \ \forall \ c \in [1, 2, 3], n \in [1, 2, \ldots, 39]]$.

### C.3 DETAILS FOR SPECTRUM ANNOTATION

When annotating an MS$^2$ mass spectrum according to its identified precursor, i.e., (peptide sequence, charge), the $m/z$ values of all possible fragment ions are calculated, where the fragment charges cannot exceed the precursor charge, and the position numbers are bounded by the length of the peptide.

The $m/z$ value, $mz$, for a fragment ion with type $t$, charge state $c$, and position number $n$ from precursor peptide sequence $S$ is

$$mz = \frac{M_{\text{fragment}} + c \cdot M_{\text{proton}}}{c},$$

where $M_{\text{fragment}}$ is the neutral mass of the frament ion and $M_{\text{proton}} = 1.0073$ Da. The neutral fragment mass depends on the ion types:

- For prefix (N-terminal) ions, a- and b-ions:

$$M_{\text{fragment}} = \sum_{i=1}^{p} M_{AA_i} + M_{\text{offset}}^t;$$

- For suffix (C-terminal) ions, y-ions:

$$M_{\text{fragment}} = \sum_{i=p+1}^{L} M_{AA_i} + M_{\text{offset}}^t,$$

where $L$ is the peptide length, $M_{AA_i}$ is the mass of amino acid at position $i$, and $M_{\text{offset}}^t$ is the ion type specific mass offset. Tables C.1 and C.2 show the amino acid masses $M_{AA}$ and ion type mass offsets $M_{\text{offset}}$.

Table C.1: Monoisotopic masses of amino acids

| Amino Acid | Single Letter | Mass (Da) |
|---|---|---|
| Alanine | A | 71.03711378471 |
| Arginine | R | 156.10111102359997 |
| Asparagine | N | 114.04292744114001 |
| Aspartic acid | D | 115.02694302383001 |
| Cysteine | C | 103.00918478471 |
| Glutamic acid | E | 129.04259308796998 |
| Glutamine | Q | 128.05857750527997 |
| Glycine | G | 57.02146372057 |
| Histidine | H | 137.05891185845002 |
| Isoleucine | I | 113.08406397713001 |
| Leucine | L | 113.08406397713001 |
| Lysine | K | 128.09496301399997 |
| Methionine | M | 131.04048491299 |
| Phenylalanine | F | 147.06841391298997 |
| Proline | P | 97.05276384885 |
| Serine | S | 87.03202840427001 |
| Threonine | T | 101.04767846841 |
| Tryptophan | W | 186.07931294985997 |
| Tyrosine | Y | 163.06332853254997 |
| Valine | V | 99.06841391299 |

Table C.2: Ion type mass offsets for fragment ion calculation

| Ion Type | Mass Offset (Da) | Description |
|---|---|---|
| b | 1.0073 | Addition of hydrogen ion ($H$) |
| a | $-26.9876$ | Loss of $CO$ and Addition of $H$ |
| y | 19.0178 | Addition of $H_2O + H$ |

Peak annotation employs a greedy assignment strategy. After computing the theoretical $m/z$ values for all applicable fragment ions, they are matched to experimental peaks using a 0.05 Th tolerance. The assignment uses a greedy strategy: for each peak in the spectrum, we identify candidate fragment ions within a 0.05 Th

mass tolerance, then assign the fragment ion with the highest probability from the global modeling among unassigned candidates. The order of fragment ions given by the global modeling is:

$$(y, 1, :) > (b, 1, :) > (y, 2, :) > (a, 1, 2) > (b, 2, :) > (y, 3) > (b, 3).$$

Peaks without valid candidates remain unannotated, and we enforce a one-to-one correspondence between peaks and fragment ions.

## D ADDITIONAL BASELINE DETAILS

We now summarize the details of our baseline models used in Section 4.1. To guarantee reproducibility, we will make the code for all experiments and the datasets utilized in this study publicly available, commencing upon the acceptance of this work.

**Linear regression model (LR).** To establish the linear regression baseline, we need to first encode each precursor $p$ into a fixed-length one-hot vector $\mathbf{x}_p$: $\mathbf{x}_p = \mathbf{x}_c \oplus \mathbf{x}_{seq}$. which concatenates two components: a one-hot encoding of the charge state $\mathbf{x}_c$ and a one-hot encoding of the peptide sequence $\mathbf{x}_{seq}$, where the latter is formed by concatenating one-hot vectors for each amino acid position. Zero padding is applied to ensure uniform vector length across all precursors. We train an independent linear regressor for each fragment ion $f$: $\hat{\mathbb{P}}_{\text{Linear}}(f|p) = \mathbf{w}_f^T \mathbf{x}_p + b_f$. Note that when training the linear regression baseline model, the loss is evaluated on *all* $(f, p)$ pairs, including invalid ones. $\mathbb{P}(f|p)$ is taken to be $-1$ on these pairs. The linear regression model is optimized using SGDRegressor from scikit-learn, due to the large amount of training samples. Here, the loss is set to 'epsilon_insensitive' with 'epsilon=0' to mimic L1 loss.

**1D convolutional neural network (ResCNN-1D).** The ResCNN-1D baseline extends the linear model with nonlinear transformations and a deeper architecture. The same one-hot encoding $\mathbf{x}_p$ is used as input. We train a 4-layer convolutional neural network with pooling in the first two layers, residual connections, and batch normalization throughout. A dropout rate of 0.05 is applied at the final layer after activation.

**Fully-connected feed-forward neural network (ResFFNN).** We use the same one-hot encoding $\mathbf{x}_p$ as input, we train a feedforward network predicting $\mathbb{P}(f|p)$ for all $f$'s simultaneously. The architecture consists of four fully-connected layers with residual connections (He et al., 2016):

$$\mathbf{h}_1 = \text{ReLU}(W_1\mathbf{x}_p + \mathbf{b}_1) \tag{D.1}$$

$$\mathbf{h}_2 = \text{ReLU}(W_2\mathbf{h}_1 + \mathbf{b}_2)/2 + \mathbf{h}_1/2 \tag{D.2}$$

$$\mathbf{h}_3 = \text{ReLU}(W_3\mathbf{h}_2 + \mathbf{b}_3)/2 + \mathbf{h}_2/2 \tag{D.3}$$

$$\hat{\mathbb{P}}_{\text{NN}}(f|p) = [\text{sigmoid}(\mathbf{W}_4\mathbf{h}_3 + \mathbf{b}_4)]_f \tag{D.4}$$

Dropout with rate 0.15 is applied after activations during training for regularization.

Both ResCNN-1D and ResFFNN minimize the L1 loss over valid fragment-precursor pairs $(f, p)$, i.e., pairs with $\pi(f, p) = 1$. For both models, we set the hidden layer size to 512 and trained for 200 epochs with a batch size of 512, and optimized using AdamW with a learning rate of $10^{-3}$.

**Transformer.** Recall that $m$ is the length of the combined input sequence. Then our transformer model is defined by

$$\mathbf{X}^{(0)} = \text{Embed}(p) + P \in \mathbb{R}^{m \times d_0}, \tag{D.5}$$

$$\mathbf{X}^{(\ell)} = \text{EncoderLayer}(\mathbf{X}^{(\ell-1)}), \qquad \ell = 1, \dots, L, \tag{D.6}$$

$$\hat{\mathbb{P}}_{\text{TF}}(f \mid p) = \text{sigmoid}(\mathbf{W}_o \mathbf{X}^{(L)} + b_o), \tag{D.7}$$

where $P$ is a positional embedding and each EncoderLayer (Dosovitskiy et al., 2020) is a standard multi-head self-attention with $H$ heads followed by a full-connected layer of width $d_{\text{ff}}$, a residual connection, and layer

normalization. Our self-attention block processes the entire input sequence concurrently, allowing each token to attend to every other token in turn to construct contextual representations. We apply a dropout rate of 0.2 inside every attention and feed-forward block. Finally, a linear output head $\mathbf{W}_o \in \mathbb{R}^{d \times d_{\text{ff}}}$ followed by a bias $b_o$ and a sigmoid activation function to get per-position probabilities. We set $d_0 = 180, H = 4, L = 4, d_{\text{ff}} = 512$. All trainings minimize the same masked L1 loss over valid fragment positions as before.

# E    ADDITIONAL EXPERIMENTAL DETAILS

## E.1    TRAINING DETAILS FOR TRANSFORMER.

- **Data loading:** Batch size $B = 1024$, shuffle for training, `num_workers`=4 (train) / 1 (evaluation).

- **Optimizer:** AdamW with initial learning rate $1 \times 10^{-3}$ and weight decay $3 \times 10^{-3}$.

- **Scheduler:** ReduceLROnPlateau (factor 0.2, patience 5 epochs, minimum LR $1 \times 10^{-6}$), stepped on validation L1 loss.

- **Loss:** Masked L1 loss over valid fragment positions, see (4.1). where $S_b$ indexes fragments with valid (theoretically possible) ions in sample $b$.

- **Training:** 100 epochs on NVIDIA GPUs (4×A100).

## E.2    COMPUTATIONAL RESOURCES

The transformer benchmark model is trained on a node with 4 NVIDIA A100 GPUs and 256 GiB memory for 4 hours per 100 epochs. The other benchmark experiments are performed on a machine with Intel® Core™ i9-13900K × 32 CPUs and 128 GiB memory. Each benchmark experiment takes up to 12 minutes, including both training and evaluation.

# F    EVALUATION DETAILS

## F.1    NORM-BASED METRICS

Here, we provide a concise summary of loss functions and evaluation metrics for the fragment ion probability vector $\boldsymbol{P}_p$ for precursor $p$. Our primary focus will be on norm-based metrics, although alternative metrics, such as divergence-based metrics and binary cross-entropy, are also available. Norm-based metrics are simple, differentiable, and widely used when the magnitude of error matters uniformly.

**Norm-based Metrics**    Except for L1 loss defined in (4.1), we also consider the following metrics between $\hat{\boldsymbol{P}}_p$ and $\boldsymbol{P}_p$:

$$\text{MSE} = \frac{\sum_{f \in \mathcal{F}} \pi(f, p)(\hat{\mathbb{P}}(f|p) - \mathbb{P}(f|p))^2}{\sum_{f \in \mathcal{F}} \pi(f, p)},$$

$$\text{SA} = 1 - \frac{2}{\pi} \arccos \frac{\langle \boldsymbol{P}_p, \hat{\boldsymbol{P}}_p \rangle}{\max\{\|\boldsymbol{P}_p\|_2 \|\hat{\boldsymbol{P}}_p\|_2, \epsilon\}}. \tag{F.1}$$

In this scenario, the value of SA is restricted to the range $[-1, 1]$. Consequently, the larger the value of SA, the more favorable the alignment exists between the predicted and the target probability vectors.

## F.2 SUPPORT-RECOVERY METRICS

First, we want to know how many ion fragment probabilities are not zero, in which case we identify the nonzero coordinates in our predictions. For each precursor, we define the support of the true probability vector $\boldsymbol{P}_p$ and the predicted probability vector $\hat{\boldsymbol{P}}_p$ as

$$S = \{f \in \mathcal{F} : \mathbb{P}(f|p) > 0\}, \hat{S} = \{f \in \mathcal{F} : \hat{\mathbb{P}}(f|p) > \tau\},$$

where $\tau > 0$ is the threshold. In our experiments, we take $\tau = 0.001$. Then, the standard evaluation metrics for evaluating the existence of fragment ions are defined as follows:

$$\text{Accuracy} = \frac{|S \cap \hat{S}| + |([d] \setminus S) \setminus \hat{S}|}{d}, \qquad \text{fraction of correct predictions;}$$

$$\text{Sensitivity} = \frac{|S \cap \hat{S}|}{|S|}, \qquad \text{fraction of true positives that are recovered;}$$

$$\text{Specificity} = \frac{|([d] \setminus S) \setminus \hat{S}|}{|[d] \setminus S|}, \qquad \text{fraction of true negatives that are correctly predicted.}$$

Here:

- $S \cap \hat{S}$ = true positives (TP),
- $\hat{S} \setminus S$ = false positives (FP),
- $S \setminus \hat{S}$ = false negatives (FN),
- $([d] \setminus S) \setminus \hat{S}$ = true negatives (TN).

## F.3 (OPTIONAL) REGRESSION CONFUSION MATRIX.

Let $\mathbb{P}(f|p), \mathbb{P}(f|p) \in [0, 1]$, be the true and predicted values at fragment ion $f$ for $f \in \mathcal{F}$. We extend the support recovery metrics to the regression confusion matrix. We partition $[0, 1]$ into ten equal intervals:

$$B_i = \begin{cases} \left[\frac{i}{10}, \frac{i+1}{10}\right), & i = 0, 1, \dots, 8, \\ \left[\frac{9}{10}, 1\right], & i = 9. \end{cases}$$

Define the bin-index function $\beta \colon [0, 1] \to \{0, 1, \dots, 9\}$ by $\beta(v) := \min\big(\lfloor 10v \rfloor, 9\big)$. The confusion matrix $\mathbf{C} \in \mathbb{N}^{10 \times 10}$ has entries

$$\mathbf{C}_{ij} = \frac{\#\{f \in \mathcal{F} : \beta(\hat{\mathbb{P}}(f|p)) = i, \ \beta(\mathbb{P}(f|p)) = j\}}{\#\{f \in \mathcal{F} : \beta(\mathbb{P}(f|p)) = j\}},$$

for $i, j = 1, \dots, 10$, where we normalize by the number of true values in the corresponding bin.

# G ADDITIONAL RESULTS

## G.1 MODEL COMPARISON ON FRAGMENTATION ION PROBABILITY PREDICTION

Table G.1 shows that across both fragment-ion and precursor-level assessments, transformer consistently achieves the lowest prediction errors and the best spectral-angle alignment, making it the most accurate model for fragment-ion probability calibration. The ResFFNN comes in as a clear runner-up, confirming that deep nets offer substantial gains over simpler methods. Linear regression provides moderate improvements above the BoF approach, which itself outperforms the naive Global baseline. In sum, there is a clear performance hierarchy, from the global model up through BoF and linear regression to the deep networks, culminating in the transformer's superior ability to model peptide fragmentation probabilities.

Table G.1: Model performance comparison on fragmentation ion probability prediction. The best results are in **bold**, the second best ones are underlined.

| Model | Fragment ion level | | | Precursor level | | |
|---|---|---|---|---|---|---|
| | **L1** | **MSE** | **SA** | **L1** | **MSE** | **SA** |
| Global | $0.2399 \pm 0.0002$ | $0.1007 \pm 0.0002$ | $0.5205 \pm 0.0007$ | $0.2437 \pm 0.0002$ | $0.0994 \pm 0.0002$ | $0.5578 \pm 0.0004$ |
| BoF | $0.1788 \pm 0.0002$ | $0.1185 \pm 0.0003$ | $0.4673 \pm 0.0009$ | $0.1788 \pm 0.0001$ | $0.1184 \pm 0.0001$ | $0.5086 \pm 0.0006$ |
| LR | $0.1293 \pm 0.0003$ | $0.0561 \pm 0.0002$ | $0.6597 \pm 0.0008$ | $0.1258 \pm 0.0002$ | $0.0540 \pm 0.0002$ | $0.6951 \pm 0.0004$ |
| ResCNN-1D | $0.0759 \pm 0.0006$ | $0.0249 \pm 0.0004$ | $0.7798 \pm 0.0021$ | $0.0720 \pm 0.0007$ | $0.0229 \pm 0.0005$ | $0.8079 \pm 0.0022$ |
| ResFFNN | $0.0739 \pm 0.0003$ | $0.0238 \pm 0.0001$ | $0.7845 \pm 0.0006$ | $0.0687 \pm 0.0002$ | $0.0213 \pm 0.0000$ | $0.8182 \pm 0.0003$ |
| Transformer | $\mathbf{0.0589 \pm 0.0002}$ | $\mathbf{0.0178 \pm 0.0000}$ | $\mathbf{0.8158 \pm 0.0005}$ | $\mathbf{0.0558 \pm 0.0002}$ | $\mathbf{0.0163 \pm 0.0000}$ | $\mathbf{0.8467 \pm 0.0002}$ |

Table G.2: Model performance comparison on fragmentation ion existence. The best results are in **bold**, the second best ones are underlined. **Acc**: accuracy; **Sen**: sensitivity; **Spec**: specificity.

| Model | Fragment ion level | | | Precursor level | | |
|---|---|---|---|---|---|---|
| | **Acc** | **Sen** | **Spec** | **Acc** | **Sen** | **Spec** |
| Global | $0.6573 \pm 0.0009$ | $\mathbf{1.0000 \pm 0.0000}$ | $0.0000 \pm 0.0000$ | $0.6993 \pm 0.0007$ | $\mathbf{1.0000 \pm 0.0000}$ | $0.0000 \pm 0.0000$ |
| BoF | $0.8141 \pm 0.0006$ | $0.3801 \pm 0.0011$ | $0.8335 \pm 0.0005$ | $0.8027 \pm 0.0008$ | $0.4435 \pm 0.0009$ | $0.7683 \pm 0.0005$ |
| LR | $0.7345 \pm 0.0043$ | $0.9105 \pm 0.0021$ | $0.3688 \pm 0.0144$ | $0.7661 \pm 0.0053$ | $0.9213 \pm 0.0021$ | $0.3771 \pm 0.0286$ |
| ResCNN-1D | $0.8621 \pm 0.0026$ | $0.8437 \pm 0.0040$ | $0.7412 \pm 0.0060$ | $0.8696 \pm 0.0023$ | $0.8640 \pm 0.0047$ | $0.7167 \pm 0.0069$ |
| ResFFNN | $0.8602 \pm 0.0018$ | $0.8529 \pm 0.0025$ | $0.7342 \pm 0.0039$ | $0.8708 \pm 0.0017$ | $0.8766 \pm 0.0026$ | $0.7150 \pm 0.0038$ |
| Transformer | $\mathbf{0.9494 \pm 0.0005}$ | $0.7672 \pm 0.0025$ | $\mathbf{0.9218 \pm 0.0008}$ | $\mathbf{0.9514 \pm 0.0006}$ | $0.8001 \pm 0.0034$ | $\mathbf{0.9189 \pm 0.0017}$ |

## G.2 CASE STUDIES ON RESFFNN

Table G.3 shows the precursors with the bottom 15 ranks for sensitivity and specificity in a test set. From the table, the presence of fragment ions is underpredicted for precursors with long peptide sequences and high charge states, including 12 out of 15 precursors with peptide lengths of $\geq 34$, as well as all precursors with a charge of 3+. The 15 precursors listed have low specificity but high sensitivity, characterized by short peptides (lengths around 10) and a charge of 1+.

The increasing number of possible fragment ions depends on the length of the peptide sequence and the precursor charge. This presents the challenge that model predictions must adequately consider the variance of possible fragment ions for each precursor input.

## G.3 EVALUATION ON FRAGMENT ION INTENSITY PREDICTION MODELS

To evaluate whether the two state-of-the-art fragment ion intensity or mass spectrum predictors, Prosit (Wilhelm et al., 2021) and AlphaPeptDeep (Zeng et al., 2022), can accurately predict fragment ion existence (a closely related but distinct task from intensity prediction), we employed the Prosit_2020_intensity_HCD and AlphaPeptDeep_ms2_generic models available through Koina (https://koina.wilhelmlab.org). We predicted spectra for precursors across all five test sets using a fixed collision energy of 30. For each predicted spectrum, we applied L2 normalization to the predicted intensities and used the same threshold $\epsilon = 10^{-6}$ s in Section 3.2 to determine fragment ion presence. We computed the precursor level accuracy, sensitivity, and specificity. The results are shown in Table G.4. In comparison to the results presented in Table G.2, both models demonstrate superior accuracy compared to BoF, albeit significantly lower than Transformer. While they exhibit high specificity, their sensitivity remains notably low.

Table G.3: Precursors with the bottom 10 precursors on sensitivity and specificity in the No.1 test set. **PID**: the precursor index; **Seq**: the precursor sequence; **#PSM**: the number of spectra identified to the precursor; **Len**: the length of the precursor sequence; **ACC**, **Sen** and **Spec**: the same as in Table G.2.

| PID | Seq | Charge | #PSMs | Len | Acc | Sen | Spec |
|---|---|---|---|---|---|---|---|
| The bottom 15 precursors on sensitivity | | | | | | | |
| 248155 | IVERPLPGYPDAEAPEPSSAGAQAAEEPSGAGSEELIK | 3 | 635 | 38 | 1.00 | 0.45 | 1.00 |
| 260783 | KGSITSVQAIYVPADDLTDPAPATTFAHLDATTVLSR | 3 | 1315 | 37 | 0.99 | 0.48 | 0.98 |
| 162400 | GAEASAASEEEAGPQATEPSTPSGPESGPTPASAEQNE | 3 | 1091 | 38 | 0.86 | 0.49 | 0.90 |
| 224171 | IFPPETSASVAATPPPSTASAPAAVNSSASADKPLSNMK | 3 | 949 | 39 | 0.95 | 0.51 | 0.94 |
| 231482 | IKQDSNLIGPEGGVLSSTVVPQVQAVFPEGALTK | 3 | 255 | 34 | 0.91 | 0.51 | 0.89 |
| 309458 | LKPAFIKPYGTVTAANSSFLTDGASAMLIMAEEK | 3 | 248 | 34 | 0.94 | 0.52 | 0.91 |
| 37623 | AQAALQAVNSVQSGNLALAASAAAVDAGMAMAGQSPVLR | 3 | 786 | 39 | 0.97 | 0.52 | 0.96 |
| 129471 | ESQPSPPAQEAGYSTLAQSYPSDLPEEPSSPQER | 3 | 169 | 34 | 0.82 | 0.52 | 0.79 |
| 146277 | FIGAGAATVGVAGSGAGIGTVFGSLIIGYAR | 3 | 3347 | 31 | 0.97 | 0.52 | 0.94 |
| 299795 | LGGGMPGLGQGPPTDAPAVDTAEQVYISSLALLK | 3 | 782 | 34 | 0.96 | 0.52 | 0.95 |
| 272819 | KPPVPLDWAEVQSQGEETNASDQQNEPQLGLK | 3 | 593 | 32 | 0.95 | 0.52 | 0.91 |
| 8860 | AEDGFEDQILIPVPAPAGGDDDYIEQTLVTVAAAGK | 3 | 463 | 36 | 0.93 | 0.52 | 0.91 |
| 564061 | VMDMLHSMGPDTVVITSSDLPSPQGSNYLIVLGSQR | 3 | 209 | 36 | 0.89 | 0.52 | 0.87 |
| 6820 | ADIDVSGPSVDTDAPDLDIEGPEGK | 3 | 881 | 25 | 0.95 | 0.53 | 0.90 |
| 93339 | EAKPGAAEPEVGVPSSLSPSSPSSSWTETDVEER | 3 | 558 | 34 | 0.96 | 0.53 | 0.94 |
| The bottom 15 precursors on Specificity | | | | | | | |
| 240926 | IQLVEEELDR | 1 | 56 | 10 | 1.00 | 1.00 | 0.00 |
| 243342 | ISEQFTAMFR | 1 | 64 | 10 | 0.95 | 1.00 | 0.00 |
| 249743 | IVVVTAGVR | 1 | 91 | 9 | 1.00 | 1.00 | 0.00 |
| 219883 | IDIIPNPQER | 1 | 73 | 10 | 0.95 | 1.00 | 0.00 |
| 90206 | DYGVLLEGSGLALR | 1 | 36 | 14 | 1.00 | 1.00 | 0.00 |
| 68977 | DITSDTSGDFR | 1 | 103 | 11 | 0.95 | 1.00 | 0.00 |
| 67159 | DIDEVSSLLR | 1 | 32 | 10 | 1.00 | 1.00 | 0.00 |
| 71929 | DLFDPIIEDR | 1 | 42 | 10 | 0.95 | 1.00 | 0.00 |
| 85797 | DTQIQLDDAVR | 1 | 57 | 11 | 0.90 | 1.00 | 0.00 |
| 600998 | YLTVAAIFR | 1 | 84 | 9 | 0.94 | 1.00 | 0.00 |
| 373876 | NLQGISSFR | 1 | 40 | 9 | 0.88 | 1.00 | 0.00 |
| 386668 | NYYEQWGK | 1 | 38 | 8 | 0.93 | 1.00 | 0.00 |
| 340892 | LVIITAGAR | 1 | 97 | 9 | 1.00 | 1.00 | 0.00 |
| 238210 | INVYYNEATGGK | 1 | 143 | 12 | 0.95 | 0.95 | 0.00 |
| 588833 | WTLLQEQGTK | 1 | 39 | 10 | 1.00 | 0.95 | 0.00 |

Table G.4: Evaluation on fragmentation ion existence predicted by fragment ion intensity models at precursor level. **Acc**: accuracy; **Sen**: sensitivity; **Spec**: specificity.

| Model | Acc | Sen | Spec |
|---|---|---|---|
| Prosit | $0.8539 \pm 0.0004$ | $0.4844 \pm 0.0005$ | $0.8291 \pm 0.0004$ |
| AlphaPeptDeep | $0.8530 \pm 0.0003$ | $0.3413 \pm 0.0004$ | $0.8912 \pm 0.0003$ |

