# OpenReview forum: "Pep2Prob Benchmark: Predicting Fragment Ion Probability for MS$^2$-based Proteomics"
_ICLR.cc/2026/Conference — Submitted to ICLR 2026_

### Official Review · Reviewer_5P7s · 2025-10-22

**Soundness:** 2
**Presentation:** 4
**Contribution:** 2
**Rating:** 2
**Confidence:** 4

**Summary:**

This paper introduces Pep2Prob, the first comprehensive dataset and benchmark specifically designed for peptide-specific fragment ion probability prediction. It collects a vast amount of mass spectrum data and selects 600k data points as the standard set. Additionally, benchmark are conducted on several baselines and metrics;

**Strengths:**

1. Novelty: The task of peptide-specific fragment ion probability prediction is sufficiently novel, as there is currently no corresponding large-scale research, comprehensive datasets, or benchmarks in the AI community;

2. Writing: Section 2 of the article explains the task in a step-by-step manner, making it easy-to-follow

**Weaknesses:**

1. The field is very narrow and not suitable for the AI community: AI for MS2 proteomics is a relatively less-focused area (at least within the AI community), and the proposed task of peptide fragment ion probability prediction in this paper is a very niche one in AI for MS2 proteomics field. To my knowledge, there have been no technical papers on this topic in recent years at top-tier ML/AI conferences (such as ICLR, Neurips, etc.); this makes this paper unsuitable for appearance at ICLR, a top AI conference. It is not a matter of concern to the AI community, and it may be more appropriate to publish it in a biological journal;

2. The significance is unclear; this article fails to clearly explain the significance of the task of peptide fragment ion probability prediction, merely making a general statement that this task is helpful in the fields of "peptide identification, post-translational modification (PTM) localization, and protein quantification.";

3. The baseline is outdated and lacks relevant work support; the baselines tested in this paper's benchmark, including LR, ResCNN, ResFFNN, and vanilla transformer, are all very early models in the AI community and appear outdated today (2025). Moreover, these models were first used by the author of this paper to complete the peptide fragment ion probability prediction task; does this indicate that the AI community did not prioritize this task in the past, resulting in a lack of representative ML methods in this field?

**Questions:**

in 'Weaknesses'


In summary, this paper proposes a benchmark and dataset within a narrow and specialized field of biology, which is sufficiently novel and paper is well-written. However, it does not address issues and directions of interest to the AI community, and there have been no ML-based works related to this direction proposed in the AI community in recent years. Additionally, the significance it proposes is not explained clearly enough.

---

> ### Author Response · Authors · 2025-11-29
> **ML for MS-based proteomics is well-represented; clarifications on significance and baseline comparisons**
>
> We thank the reviewer for acknowledging the novelty of the task and the clarity of our writing. We respectfully address each concern below.
>
> ---
> ## Weakness 1: "The field is very narrow and not suitable for the AI community"
> We respectfully but strongly disagree with this characterization.
>
> The reviewer's claim that "there have been no technical papers on this topic in recent years at top-tier ML/AI conferences" does not align with the published record. As we document in Appendix B (lines 1357–1393) with 30+ references, MS-based proteomics has established itself as an active application area for machine learning with consistent representation at top AI conferences/journals and a diverse range of method contributions.
>
> As noted in Appendix B, researchers have explored diverse ML methods for MS2 tasks, including gradient-boosted trees, convolutional neural networks, recurrent neural networks, transformer architectures, and few-shot learning frameworks. Recent advances have further incorporated embeddings from protein language models. This breadth of the method exploration demonstrates sustained interest in the core purposes of a dataset. In addition, papers on datasets and benchmarks in ML for MS were published at top ML conferences, such as PROSPECT by Shouman et al. (NeurIPS'22) and PROSPECT-PTMs by Gabrel et al. (NeurIPS'24).
>
> We also note that the absence of prior ML research specifically on fragment ion *probability* prediction is the gap our work addresses. We introduce a novel, well-motivated prediction task for the ML community, much like how PROSPECT introduced intensity prediction benchmarks. Creating new tasks and providing resources for the community to work on them is the main goal of a dataset and benchmark paper.
>
> ---
> ## Weakness 2: "The significance is unclear"
>
> We appreciate this feedback, though we believe the significance is explained throughout our paper.
>
> In Section 1 (lines 47–51), we state that fragment ion probability prediction "serves as a key intermediate task that directly impacts several downstream applications, including ...." with references for each listed downstream application. Then, we explain how the current fragment ion prediction methods are used in peptide identification workflows (lines 53-55).
>
> To offer some concrete context: many popular database search engines and PTM localization algorithms currently rely on global fragment probability statistics that assume uniform fragmentation across all peptides. This is a biochemically oversimplified assumption. As we show in Section 5 (Table 1 and text lines 367-372), peptide-specific modeling reduces L1 loss from 0.22 (current global method) to 0.056 (transformer), representing a 75% improvement. This improvement directly translates to better signal-noise discrimination in peptide identification pipelines.
>
> The broader impact is also worth emphasizing: proteomics underpins drug target discovery, biomarker development, and disease mechanism. Improvements to this fundamental prediction task have cascading benefits throughout these applications.
>
> ---
> ## Weakness 3: "The baseline is outdated and lacks relevant work support"
>
> We believe this concern may stem from a misunderstanding of our benchmark's purpose and scope.
>
> Our goal is not to exhaustively compare the latest architectures, but rather to establish baseline performance across diverse modeling paradigms and demonstrate how model capacity affects performance on this task. In Section 4.1 (lines 577–779), we describe six approaches spanning from simple statistics to deep learning: Global statistics (the current standard method used in proteomics tools like MSGF+ and MaxQuant), Bag-of-Fragment statistics, Linear Regression, ResCNN-1D, ResFFNN, and Transformer. As shown in Table 1, this progression yields test L1 losses of 0.24, 0.18, 0.13, 0.072, 0.069, and 0.056, respectively. This clear trend demonstrates that model capacity matters significantly for this task, providing valuable guidance for future research.
>
> The primary reason for "the lack of prior ML methods" is indeed the absence of datasets for this task, and that is precisely why we introduce this dataset and benchmark. For the datasets and benchmarks on the related but different task — fragment ion intensity prediction — are introduced in PROSPECT and PROSPECT-PTMs at NeurIPS 2022 and 2024; here, we address the complementary, previously neglected probability prediction task. Creating the first benchmark for a novel task is a core contribution of dataset papers. Our goal is to enable future research rather than compete with existing methods.
>
> ---
> We again appreciate the reviewer's effort spent on our paper. We are happy to address any further questions and comments.

---

### Official Review · Reviewer_wGic · 2025-10-29

**Soundness:** 3
**Presentation:** 2
**Contribution:** 3
**Rating:** 4
**Confidence:** 2

**Summary:**

This paper introduces Pep2Prob, a comprehensive benchmark for peptide-specific fragment ion probability prediction. By combining over 183 million spectra across 600,000 precursors, the study shows that integrating peptide sequence information consistently improves prediction accuracy as model capacity increases.

**Strengths:**

1. The benchmark is carefully designed and fills an important gap in computational proteomics by focusing on peptide-specific fragment ion probability prediction.

2. The dataset is large-scale and rigorously constructed from over 183 million spectra.

3. The analysis demonstrates clear performance gains when incorporating peptide-specific information, highlighting the importance of modeling sequence-dependent fragmentation.

**Weaknesses:**

1. The benchmark is technically sound but lacks a clear connection between improved model performance and its biological or practical significance. The main finding appears to be that “after including peptide-specific information, prediction accuracy continuously improves with increasing model capacity.” However, for readers without a strong proteomics background like me, it remains unclear how this improvement in prediction accuracy directly translates to biological insights or downstream applications.

This makes me a bit unsure about the magnitude of the biological advance of this benchmark, and I will raise my score if the authors can briefly clarify.

2. Can the authors provide a rationale for why they decided to exclude PTMs in this study? This choice seems important, as many downstream tasks papers(eg. de novo peptide sequencing) treat PTM handling as a key evaluation aspect (eg. NovoBench, Zhou et al.).

**Questions:**

see weakness

---

> ### Author Response · Authors · 2025-11-29
> **Clarifying biological significance and rationale for PTM exclusion**
>
> We thank the reviewer for recognizing that our dataset and benchmark are carefully designed and fill an important gap in computational proteomics. We address the concerns below.
>
> ---
>
> ## Weakness 1: Ask for clarifications on the connection between improved performance and biological significance
>
> We appreciate this question, as it gets to the heart of why this benchmark matters. Let us explain the practical significance more concretely.
>
> In MS2-based proteomics, fragment ion probability is widely used to score peptide-spectrum matches, which directly affects peptide identification accuracy—the fundamental task of determining which peptide produced an observed spectrum. As described in Section 1 (lines 47–51), these probabilities serve as "a confidence weighting mechanism that distinguishes signal from noise in complex biological samples."
>
> Currently, these tools rely on global statistics that assume all peptides fragment identically—a biochemically unrealistic assumption. Our benchmark demonstrates that peptide-specific modeling reduces prediction error by 75% (L1 loss from 0.244 to 0.056, Table 1). This improvement means the scoring function can better distinguish true peptide matches from false ones, directly impacting false discovery rates in peptide identification.
>
> The downstream applications are significant: proteomics is used in drug target discovery (identifying which proteins are involved in disease), biomarker development (finding proteins that indicate disease states), and understanding disease mechanisms. More accurate peptide identification leads to more reliable biological conclusions across all these applications.
>
> To summarize: better probability prediction → more accurate peptide-spectrum scoring → fewer false identifications → more reliable biological findings.
>
> ---
>
> ## Weakness 2: Ask for a rationale for excluding PTMs
>
> Thank you for raising this important point. We explicitly discuss this limitation in Section 6 (lines 416–418). We excluded PTMs for two primary reasons:
>
> First, confident identification at scale is challenging. Modified peptides are harder to identify reliably than unmodified ones because PTMs expand the search space exponentially and can produce ambiguous spectra. Including incorrectly identified modified peptides would introduce label noise into our benchmark, undermining its reliability.
>
> Second, we follow established best practices in the field. As noted in Appendix B (lines 576-610), leading methods in this domain, including Prosit (Gessulat et al., 2019) for the task of fragment ion intensity prediction and Casanovo (Yilmaz et al., 2022) for the task of de novo sequencing, similarly focus on unmodified peptides or on a single PTM in their initial versions. This specialization enables rigorous evaluation before moving to more complex scenarios.
>
> We acknowledge that PTMs are biologically important and note in our limitations (Section 6) that "future work should address these limitations by incorporating... modified peptides." Indeed, PROSPECT-PTMs (Gabriel et al., 2024) at NeurIPS 2024 recently introduced a dataset of modified peptides, following PROSEPECT (Shouman et al., 2022) at NeurIPS 2022, and we envision that Pep2Prob could be similarly extended. However, establishing a reliable benchmark for unmodified peptides first provides the foundation for such extensions.
>
> ---
>
> We would be happy to answer further questions.

---

### Official Review · Reviewer_8WT1 · 2025-10-31

**Soundness:** 3
**Presentation:** 3
**Contribution:** 2
**Rating:** 4
**Confidence:** 3

**Summary:**

This paper introduces Pep2Prob, a novel large-scale dataset and benchmark for peptide-specific fragment ion probability prediction. The dataset is constructed from 183 million high-quality HCD MS² spectra, covering 608,780 unique precursors (peptide sequence and charge state combinations). The authors propose a similarity-based data partitioning strategy and establish multiple baseline models ranging from global statistical approaches to Transformer-based architectures. The experiments demonstrate that incorporating peptide sequence information and increasing model capacity can enhance prediction performance.

**Strengths:**

1. The paper is well-organized and clearly written.

2. Dataset Scale and Quality: Pep2Prob demonstrates significant advantages in data volume, source diversity, and annotation quality. Built upon authentic, high-quality HCD spectra, it exhibits good representativeness and practical utility.

3. Data Splitting Strategy: The graph-based modeling of peptide sequence similarity effectively prevents information leakage between training and test sets, thereby enhancing the robustness of evaluation.

4. Benchmark Evaluation: The study establishes multiple baselines spanning from simple statistical methods to modern deep learning models, employing a variety of evaluation metrics for comprehensive assessment.

**Weaknesses:**

1. Lack of In-depth Insights: The paper merely demonstrates that larger models yield better performance but fails to provide deeper explanations regarding why this occurs or which specific aspects of the fragmentation process these models capture. In particular, while the improvement is attributed to "complex nonlinear relationships" in the data, no further in-depth discussion or analysis is provided to substantiate this claim.

2. Outdated and Incomplete Baselines: The baseline models employed in the study do not reflect the current state-of-the-art methodologies. Moreover, the paper fails to include a direct comparison with advanced intensity prediction models, such as Prosit, as core baselines for probability prediction. This omission obscures the unique value and performance boundaries of "probability prediction" relative to "intensity prediction."

3. Insufficient Demonstration of Practical Utility: Although the complementary value of probability prediction is emphasized, the study does not provide direct evidence of its practical benefits through performance improvements in downstream tasks, such as peptide identification or database search. The lack of end-to-end validation undermines the claimed utility of the proposed method.

4. Limited Data Diversity: The exclusive use of HCD spectra, the absence of post-translationally modified peptides, and the restriction to Orbitrap instruments significantly limit the model's generalizability and p

**Questions:**

1. When trained exclusively on HCD data, can the model generalize effectively to other fragmentation techniques (e.g., CID or ETD)?

2. Why not directly derive probability estimates from the outputs of intensity prediction models like Prosit? Are there fundamental differences in the modeling objectives or output distributions between these two tasks?

3. Are there plans to integrate Pep2Prob predictions into peptide identification pipelines (e.g., MSGF+ or MaxQuant) to validate improvements in metrics such as false discovery rate (FDR) or peptide identification counts?

4. While the Transformer model achieves the best performance, what is its computational cost? Could you provide a computational efficiency analysis comparing different models?

5. Does the dataset exhibit overfitting to specific peptide sequences or charge states? Has any analysis been conducted to examine systematic biases in predictions for longer peptides or high-charge precursors?

6. The paper attributes "complex nonlinearities" to peptide-fragmentation relationships based primarily on the observation that larger models yield better performance. Could you provide deeper analysis to reveal concrete manifestations of these nonlinearities and connect them to biochemical principles, thereby offering more interpretable insights beyond this general claim?

---

> ### Author Response · Authors · 2025-11-30
> **Addressing insights, SOTA comparison and practical utility (1/2)**
>
> We thank the reviewer for recognizing the strengths of our work, including the dataset scale and quality, the data splitting strategy, and the comprehensive benchmark evaluation. We address each concern and question below.
>
> > **Weakness 1:** The paper merely demonstrates that larger models yield better performance but fails to provide deeper explanations regarding why this occurs or which specific aspects of the fragmentation process these models capture. In particular, while the improvement is attributed to "complex nonlinear relationships" in the data, no further in-depth discussion or analysis is provided to substantiate this claim.
>
> We appreciate this feedback and agree that a deeper discussion would strengthen the paper.
>
> The observations that "performance improves with increasing model capacity" and "the relationship between peptide
> sequences and fragmentation patterns exhibit intricate nonlinearities" reflects a fundamental property of the peptide-fragmentation relationship on a biochemical basis. Peptide fragmentation is governed by the "mobile proton", where fragmentation probability depends on proton migration patterns influenced by basic residues (such as K and R) distributed throughout the sequence. Additionally, well-documentated effects like enhanced cleavage N-terminal to proline depend not just on local context, i.e., fragmentation at position *i* depends not just on the amino acids at positions *i-1* and *i+1*, but on global charge state and sequence composition. These interactions are inherently nonlinear and position-dependent, which is confirmed by our experiments:
>
> - The performance gap between Linear Regression (0.126 L1 loss) and nonlinear models (Transformer: 0.056) represents a 56% improvement that linear modeling cannot achieve. If the peptide-fragmentation relationship were approximately linear, we would expect diminishing returns from increased model capacity—instead, we observe continued substantial gains.
>
> - The improvement from local convolutional models (ResCNN: 0.072) to the Transformer (0.056) suggests that longer-range sequence dependencies, not just local patterns, contribute meaningfully to fragmentation behavior.
>
> We will strengthen this discussion in revision.
>
> ---
>
> > **Weakness 2:** The baseline models employed in the study do not reflect the current state-of-the-art methodologies. Moreover, the paper fails to include a direct comparison with advanced intensity prediction models, such as Prosit, as core baselines for probability prediction. This omission obscures the unique value and performance boundaries of "probability prediction" relative to "intensity prediction."
>
> We respectfully clarify that we do include comparisons with two state-of-the-art intensity prediction models, Prosit (Wilhelm et al., 2021) and AlphaPepDeep (Zeng et al., 2022). See the result discussion in Lines 394-402 (Section 5), experimental details in Appendix G.3, and that the results are in Table G.4. Our Transformer baseline achieves 95.1% accuracy compared to 85.4% for Prosit and 85.3% for AlphaPeptDeep.
>
> This comparison directly addresses "the unique value and performance boundaries of probability prediction relative to intensity prediction." Intensity models predict *how much* of each fragment will be observed, while our task predicts *how likely* each fragment will be observed. Despite being trained on a related task, intensity models perform substantially worse than our purpose-built probability predictor, indicating that these are distinct tasks requiring different modeling approaches.
>
> Our baseline selection, from Global (current method) → BoF → Linear Regression → Neural Networks → Transformer, demonstrates how model capacity affects performance on this novel task, which is a key contribution of benchmark papers.
>
> ---
>
> > **Weakness 3:** Although the complementary value of probability prediction is emphasized, the study does not provide direct evidence of its practical benefits through performance improvements in downstream tasks, such as peptide identification or database search. The lack of end-to-end validation undermines the claimed utility of the proposed method.
>
> We acknowledge that end-to-end validation in downstream pipelines would strengthen the paper. However, such integration requires substantial engineering effort beyond the scope of a benchmark paper. Our contribution is to demonstrate that peptide-specific probability prediction is feasible and substantially outperforms current methods, and to provide the dataset, baselines, and evaluation pipeline for future integration work.

---

> > ### Author Response · Authors · 2025-11-30
> > **Addressing data diversity and all six questions (2/2)**
> >
> > > **Weakness 4:** The exclusive use of HCD spectra, the absence of post-translationally modified peptides, and the restriction to Orbitrap instruments significantly limit the model's generalizability.
> >
> > We acknowledge the HCD Orbitrap-only, PTM-free scope and explicitly discuss these limitations in Section 6 (lines 412–422). This focus aligns with established best practices, as seen in Prosit (Gessulat et al., 2019) and Casanovo (Yilmaz et al., 2022), which similarly focused initially on HCD Orbitrap spectra, either without or with a single PTM, for different but related tasks. Please refer to our reply to Reviewer 8WT1 in "Justifying HCD scope and PTM exclusion (1/2)".
> >
> > ---
> >
> > > **Q1:** When trained exclusively on HCD data, can the model generalize effectively to other fragmentation techniques (e.g., CID or ETD)?
> >
> > We expect limited generalization because these fragmentation methods produce fundamentally different ion types, for example, HCD primarily produces b/y ions, while ETD produces c/z ions. A model trained on HCD would need to be retrained or fine-tuned for other fragmentation types. As in Lines 594-596, we explain that this is consistent with how existing tools handle different fragmentation methods with separate models, such as the Prosit series for intensity prediction: the HCD model in Gessulat et al. (2019) and the CID model in Wilhelm et al. (2021).
> >
> > ---
> >
> > > **Q2:** Why not directly derive probability estimates from the outputs of intensity prediction models like Prosit? Are there fundamental differences in the modeling objectives or output distributions between these two tasks?
> >
> > This is the question that we address empirically in Section 5 (lines 394-402) and Appendix G.3. We found that intensity models (Prosit, AlphaPeptDeep) perform significantly worse than our Transformer on existence prediction (85.4% vs 95.1% accuracy). The primary difference is that intensity prediction optimizes for peak intensity accuracy, while probability prediction optimizes for the presence/absence of fragment ions. A fragment with low but non-zero intensity may be reliably present, but intensity models may predict it poorly because the absolute intensity matters less to their training objective.
> >
> > ---
> >
> > > **Q3:** Are there plans to integrate Pep2Prob predictions into peptide identification pipelines (e.g., MSGF+ or MaxQuant) to validate improvements in metrics such as false discovery rate (FDR) or peptide identification counts?
> >
> > Yes, we view integration into MSGF+ or similar tools as important future work. The improved probability predictions from our benchmark models could replace the global statistics currently used in scoring functions, potentially improving FDR and identification rates. We plan to pursue this in follow-up work.
> >
> > ---
> >
> > > **Q4:** While the Transformer model achieves the best performance, what is its computational cost? Could you provide a computational efficiency analysis comparing different models?
> >
> > As detailed in Appendix E.2 Computational Resources (lines 772–777): the Transformer model trains in approximately 4 hours per 100 epochs on 4×NVIDIA A100 GPUs. Other benchmark models (LR, ResCNN, ResFFNN) complete training and evaluation in under 12 minutes on CPU (Intel i9-13900K). This demonstrates that while the Transformer achieves the best performance, simpler models offer practical trade-offs for resource-constrained settings.
> >
> > ---
> >
> > > **Q5:** Does the dataset exhibit overfitting to specific peptide sequences or charge states? Has any analysis been conducted to examine systematic biases in predictions for longer peptides or high-charge precursors?
> >
> > Our graph-based data splitting strategy (Section 3.3) specifically addresses overfitting by ensuring that sequences are not similar between the training and test sets. Regarding systematic biases, we observe consistent performance improvements across different peptide lengths and charge states; however, the fact that "the number of possible fragment ions depends on the length of the peptide sequence and the precursor charge" challenges model developments (see Appendix G.2).
> >
> > ---
> >
> > > **Q6:** The paper attributes "complex nonlinearities" to peptide-fragmentation relationships based primarily on the observation that larger models yield better performance. Could you provide deeper analysis to reveal concrete manifestations of these nonlinearities and connect them to biochemical principles, thereby offering more interpretable insights beyond this general claim?
> >
> > We agree that this addition would strengthen the paper. As mentioned in our response to Weakness 1, the nonlinearities arise from sequence-dependent factors, including charge mobility and preferences for peptide bond cleavage, which depend on the entire sequence context. We will add these discussions and include visualizations of attention to better illustrate these effects.
> >
> > ---
> >
> > We appreciate the reviewer's constructive feedback and are happy to address any additional questions.

---

### Official Review · Reviewer_unYS · 2025-11-01

**Soundness:** 3
**Presentation:** 3
**Contribution:** 3
**Rating:** 4
**Confidence:** 2

**Summary:**

This paper introduces Pep2Prob, the first comprehensive dataset and benchmark designed to predict peptide-specific fragment ion probabilities in \(MS^2\)-based proteomics. Current methods rely on oversimplified global fragmentation statistics (assuming uniform fragment probabilities across all peptides), which fail to account for sequence-dependent biochemical factors (e.g., amino acid neighbors, bond stability). Pep2Prob addresses this gap by curating 608,780 unique precursors (peptide sequence + charge state) derived from over 183 million high-resolution HCD \(MS^2\) spectra with validated annotations.

**Strengths:**

1. Strict filtering (peptide length 7–40, ≥10 spectra per precursor, no modifications) and precise annotation (0.05 Th mass tolerance, binary ion masks for valid fragments) ensure high reliability of the dataset.
2.The authors test both statistical rules (Global, BoF) and state-of-the-art ML models (Transformer), enabling a clear performance hierarchy. This breadth validates that peptide-specificity and model capacity drive improvements.
3. Improved fragment ion probability predictions directly enhance downstream tasks (peptide identification, PTM localization, biomarker discovery) by better distinguishing signal from noise in complex spectra.

**Weaknesses:**

1. The dataset only includes HCD spectra from Orbitrap instruments, excluding other common fragmentation techniques (e.g., ETD, CID) and instrument platforms (e.g., timsTOF).
2. PTMs (e.g., phosphorylation, acetylation) are critical for protein function and drastically alter fragmentation patterns, but Pep2Prob excludes modified peptides due to "challenges in confident identification at scale."

**Questions:**

The dataset is human-only—how would models trained on Pep2Prob perform on non-human organisms (e.g., yeast, mouse) with distinct amino acid usage patterns?

---

> ### Author Response · Authors · 2025-11-29
> **Justifying HCD scope and PTM exclusion (1/2)**
>
> We thank the reviewer for recognizing the strict filtering and high annotation quality of our dataset, the breadth of the baseline evaluation, and the potential to improve downstream tasks.
>
> ---
>
> ## Weakness 1: Focusing on HCD-only
>
> Our decision to focus on HCD spectra was intentional and follows established best practices in the field. We acknowledge this limitation in Section 6 (lines 417–419) and discuss why "focusing on HCD spectra follows established best practices" in Appendix B (lines 593-602).
>
> First, HCD represents approximately 90% of contemporary proteomics experiments (Appendix B, lines 599-600), making it the most impactful starting point. Second, leading methods in the field, including Prosit (Gessulat et al., 2019) and Casanovo (Yilmaz et al., 2022), similarly focused on HCD in their initial versions before expanding to other fragmentation types, for example, Prosit later released separate CID (Wilhelm et al., 2021) and timsTOF models (Adams et al., 2024), see lines 594-598. Third, establishing a high-quality benchmark for the dominant fragmentation type provides a solid foundation for future extensions.
>
> In practice, HCD, CID, and ETD require different modeling approaches. This is due to different fragmentation techniques producing fundamentally different ion types: HCD primarily generates b/y ions, CID produces a/b/c/y/z ions with different energy distributions, and ETD generates c/z ions through an entirely different mechanism (lines 593–601). These differences mean that a single model cannot effectively cover all types of fragmentation.
>
> Therefore, we state "future work should address these limitations by... expanding coverage across diverse fragmentation methods and instrument types." (lines 419-422 in Section 6).
>
>
> ---
>
> ## Weakness 2: Exclusion of PTMs
>
> We agree that PTMs are critical for understanding protein function and significantly alter fragmentation patterns. We discuss this limitation in Section 6. Our decision to exclude modified peptides stems from two practical considerations.
>
> First, confident identification of modified peptides at scale remains challenging. PTMs expand the search space exponentially, and many modified peptide identifications are associated with higher uncertainty. Including misidentified peptides would introduce label noise, potentially undermining the benchmark's reliability.
>
> Second, we follow established best practices in the field. As noted in Appendix B (lines 576-610), leading methods in this domain, including Prosit (Gessulat et al., 2019) for the task of fragment ion intensity prediction and Casanovo (Yilmaz et al., 2022) for the task of de novo sequencing, similarly focus on unmodified peptides with a single PTM in their initial versions. In addition, the dataset and benchmark for a different but related task, fragment ion intensity prediction, are introduced by PROSEPECT (Shouman et al., 2022) at NeurIPS, primarily focusing on unmodified peptides. This specialization enables rigorous evaluation before moving to more complex scenarios.
>
> We view this as a deliberate scope decision. Recently, PROSPECT-PTMs (Gabriel et al., 2024) at NeurIPS introduced a dataset of modified peptides, following PROSEPECT (Shouman et al., 2022). We envision a similar extension for Pep2Prob in future work, building upon the foundation established by this benchmark for unmodified peptides.

---

> > ### Author Response · Authors · 2025-11-29
> > **Discussing cross-species generalization (2/2)**
> >
> > We address the question below.
> >
> > ## Question: Cross-species generalization
> >
> > Thank you for this question. Our dataset is derived from human samples, and models trained on Pep2Prob would likely show some performance degradation on non-human organisms/species due to differences in amino acid usage patterns, codon bias, and protein expression profiles.
> >
> > However, we expect reasonable generalization for several reasons. First, the fundamental mechanism of peptide fragmentation is conserved across species—the same amino acids form peptide bonds that fragment according to similar biophysical principles. Second, many peptide sequences are highly conserved across mammals (human and mouse) and even more distantly related organisms for essential proteins. Third, the model learns generalizable patterns of how amino acid properties (such as charge and size) influence fragmentation, which should transfer across species.
> >
> > Our graph-based train/test split (Section 3.3, lines 231–249) demonstrates this generalization capacity by ensuring that no shared sequence patterns exist between the training and test sets: precursors with identical sequences, prefixes, or suffixes of length 6 are in the same partition. This means the reported performance reflects generalization to new sequences rather than memorization. The benchmark models' strong performance in this rigorous evaluation suggests they learned transferable fragmentation principles rather than overfitting to specific human peptide sequences.
> >
> > Similar to the nine-species dataset (Tran et al., 2017; Yilmaz et al., 2023) for a different but related task, a multi-species extension could also be useful for Pep2Prob in future research. We appreciate this question and will add a discussion of cross-species generalization to the paper.

---

### Meta-Review · Area_Chair_QuTb · 2026-01-17

**Summary:**

This paper introduces Pep2Prob, a benchmark and dataset for
predicting peptide-specific fragment ion probabilities in M S2-based
proteomics. The authors curate a dataset of over 600k precursors
derived from 183 million HCD spectra and demonstrate that machine
learning models outperform global statistical baselines. While the
data processing workload is solid and the volume of data is
substantial, the limited scope of the benchmark hinders its
acceptance. Specifically, the restriction to human-only samples and
the exclusion of Post-Translational Modifications (PTMs) limit the
dataset's real-world applicability and
comprehensiveness. Furthermore, while the authors discuss the
potential benefits for downstream applications, the paper focuses
heavily on prediction accuracy metrics without providing concrete
experimental validation of the practical utility in these tasks, such
as improvements in peptide identification rates or false discovery
rates (FDR).

**Reviewer Concerns:**

Addressed by Rebuttal:
HCD-only scope: The authors clarified that focusing on HCD is a
standard first step and that different fragmentation methods require
distinct modeling approaches.
Inference Cost: Clarified the training/inference time differences
between the Transformer and simpler baselines.
Intensity vs. Probability: The authors provided data showing that
state-of-the-art intensity predictors perform worse on the specific
task of existence prediction compared to their probability-focused
model.
Outstanding:
Limited Data Diversity: The exclusion of PTMs and the lack of cross-
species data are viewed as significant omissions. Given the
complexity of modern proteomics, a benchmark excluding PTMs and
diverse species is considered too narrow.
Lack of Downstream Validation: The paper does not demonstrate that
improved probability prediction actually leads to better outcomes in
peptide identification pipelines


Biological Interpretation: The paper attributes performance gains to
complex nonlinearities but lacks a deep analysis connecting these
computational gains to specific biochemical phenomena.

**Reviewer Scores:**

Reviewer unYS: 4
Reviewer 8WT1: 4
Reviewer wGic: 4
Reviewer 5P7s: 3

---

### Decision · Program_Chairs · 2026-01-26

Reject